# Pilot Study of the Effects of Polyphenols from Chestnut Involucre on Methane Production, Volatile Fatty Acids, and Ammonia Concentration during In Vitro Rumen Fermentation

**DOI:** 10.3390/ani11010108

**Published:** 2021-01-07

**Authors:** Yichong Wang, Sijiong Yu, Yang Li, Shuang Zhang, Xiaolong Qi, Kaijun Guo, Yong Guo, Riccardo Fortina

**Affiliations:** 1College of Animal Science and Technology, Beijing University of Agriculture, Beijing 102206, China; wangyichong200811@163.com (Y.W.); vittoriog@126.com (S.Y.); 18730431956@163.com (Y.L.); selenazh920@126.com (S.Z.); bbcqxl@126.com (X.Q.); y63guo@126.com (Y.G.); 2Dipartimento di Scienze Agrarie, Forestali e Alimentari, Università degli Studi di Torino, Largo Paolo Braccini 2, Grugliasco, TO 10095, Italy

**Keywords:** polyphenols of *Castanea mollissima *Blume involucres, in vitro fermentation, ammonia nitrogen, methane yield, acetic/propionic ratio

## Abstract

**Simple Summary:**

In China, chestnut involucres, an agricultural byproduct, are not appropriately used and are regarded as agricultural waste. To reduce the environmental pollution caused by burning or misuse of chestnut involucres, the authors extracted polyphenols from the involucres of *Castanea mollissima* Blume (PICB) to produce a new additive. PICB is known to have beneficial effects on monogastric animals. However, the effects of PICB on ruminants need further investigation. This study shows that the addition of PICB to the ruminant diet can significantly reduce methane yields and the ratio of acetic acid to propionic acid, thus improving feed efficiency. The results contribute to the knowledge of tannins from a specific part of the *Castanea mollissima* Blume plant and provide a theoretical basis for the potential use of involucres. Such an application would limit environmental pollution caused by their disposal and reduce the negative influence of ruminants on the environment.

**Abstract:**

Nutritional strategies can be employed to mitigate greenhouse emissions from ruminants. This article investigates the effects of polyphenols extracted from the involucres of *Castanea mollissima* Blume (PICB) on in vitro rumen fermentation. Three healthy Angus bulls (350 ± 50 kg), with permanent rumen fistula, were used as the donors of rumen fluids. A basic diet was supplemented with five doses of PICB (0%–0.5% dry matter (DM)), replicated thrice for each dose. Volatile fatty acids (VFAs), ammonia nitrogen concentration (NH_3_-N), and methane (CH_4_) yield were measured after 24 h of in vitro fermentation, and gas production was monitored for 96 h. The trial was carried out over three runs. The results showed that the addition of PICB significantly reduced NH_3_-N (*p* < 0.05) compared to control. The 0.1%–0.4% PICB significantly decreased acetic acid content (*p* < 0.05). Addition of 0.2% and 0.3% PICB significantly increased the propionic acid content (*p* < 0.05) and reduced the acetic acid/propionic acid ratio, CH_4_ content, and yield (*p* < 0.05). A highly significant quadratic response was shown, with increasing PICB levels for all the parameters abovementioned (*p* < 0.01). The increases in PICB concentration resulted in a highly significant linear and quadratic response by 96-h dynamic fermentation parameters (*p* < 0.01). Our results indicate that 0.2% PICB had the best effect on in-vitro rumen fermentation efficiency and reduced greenhouse gas production.

## 1. Introduction

Greenhouse gas (GHG) emissions from animal husbandry must be reduced to mitigate the effects of climate change. Anaerobic rumen fermentation produces a large amount of methane (CH_4_), accounting for 39% of GHG emissions from the animal husbandry industry [1,2]. Furthermore, in ruminants, the rate of utilization of nitrogen is slow and equal to approximately 25% of nitrogen intake [3]. A large amount of nitrogen is lost through feces and urine, causing the emission of ammonia nitrogen (NH_3_-N) and nitrous oxide (N_2_O) [4]. These losses are responsible for environmental pollution, compound the greenhouse effect, and increase the production costs of the farm. Many studies aimed at reducing GHG emissions have suggested that the addition of tannins in the diet of ruminants is one of the most promising nutritional strategies because these molecules can increase nitrogen retention and energy utilization by reducing CH_4_ and NH_3_ emissions [5].

Tannins are high molecular weight polyphenols. They are an important class of plant secondary metabolites and encompass a wide variety of molecules of different size and complexity [6]. Based on their molecular structure, they can be divided into hydrolyzable tannins (HTs) or condensed tannins (CTs) [7]. Both HTs and CTs have been shown to reduce methane production. Tannins derived from chestnut, myrobalan, mimosa, and quebracho reduced CH_4_ production in in-vitro experiments [8]. Deaville et al. [9] reported chestnut wood tannins could change the nitrogen excretion pattern, reducing nitrogen in urine while increasing nitrogen in feces.

According to the United States Food and Agriculture Organization, the worldwide production of American, European, and Chinese chestnuts and their hybrids is about 2.3 million tons per year [10]. Chestnut involucres are a byproduct of the agro-food industry, accounting for about 10%–25% of the total weight of the chestnuts [11]. These involucres are often a wasted resource, directly discarded or burned [12], producing dioxin-like toxic substances and causing environmental pollution [13]. Polyphenols extracted from the involucres of Chinese chestnut (*Castanea mollissima* Blume; PICB) mainly contain HTs such as ellagitannin and gallotannin [14]. Dong et al. [15] and Xiong et al. [16] reported the antioxidant activity of PICB from in vitro and in vivo experiments and investigated the effect of PICB on broilers. Although many studies have reported on the use of chestnut tannin from the wood of *Castanea sativa*, no literature report has addressed the effects of PICB on rumen fermentation. Aboagye and Beauchemin [17] indicated the effects of dietary polyphenols on intake, digestibility, rumen fermentation, CH_4_ production, and animal performance vary depending on diet composition, tannin source and type, dose, and molecular weight, part of the plant, stage of growth, and growing conditions. The effects of different tannin sources should be studied case by case. The aim of this study is to investigate the effects of different concentrations of PICB on pH, NH_3_-N concentration, volatile fatty acids (VFAs) and its components, gas production, and the dry matter degradation rate (IVDMD) during in vitro fermentation.

## 2. Materials and Methods

The experimental procedure was evaluated and approved by the Committee for the Care and Use of Experimental Animals, Beijing University of Agriculture (Approval No. BUA 1901030). The animals were cared for and handled according to the welfare and ethics standards for experimental animals of Beijing Municipality (Announcement No. 31 of the Standing Committee of the Beijing Municipal People’s Congress on regulations of Beijing Municipality on the administration of laboratory animals).

### 2.1. PICB Extraction

*Castanea mollissima* Blume involucres were sampled from Huairou Chestnut Station. PICB were extracted according to the method described by Shi et al. [14]. Briefly, the involucres were dried at room temperature and ground to pass through a No. 80 mesh sieve. The powder was added to 40% aqueous ethanol and extracted at 67 °C for 38 min. The supernatant was isolated and centrifuged at 4919.2× *g* for 15 min. PICB extract was obtained by drying the supernatant with a rotary evaporator. The final PICB had 52.5% total polyphenol content, among which 18.65% was ellagic acid, which is considered the effective component. The nonpolyphenol content of the PICB extract was made up of crude fiber, ash, and water [14,15].

All chemicals and reagents used in this trial were of analytical grade and were purchased from commercial sources.

### 2.2. Donor Animals, Diet, and Rumen Fluid Collection

The trial was carried out at the experimental farm of the Beijing University of Agriculture. Three healthy Angus bulls (350 ± 50 kg), equipped with permanent rumen fistula, were selected as rumen fluid donors. The cattle were raised in a confined barn, and their total mixed ration (TMR) is shown in Table 1. The cattle were fed twice daily at 08:00 and 16:00 and had free access to fresh water; the bedding was cleaned daily.

After 14 days of adaptation, samples of rumen fluid were collected from the three donors via the rumen fistula before the morning feed. Rumen fluid was then placed in a thermos flask and brought to the laboratory within 20 min. Rumen fluid was filtered through four layers of gauze into a flask while continuously flushing with CO_2_ to maintain anaerobic conditions. The operation was completed in the shortest possible time.

### 2.3. Experimental Design and Treatments

The trial was carried out three times. The basic fermentation substrate for the in-vitro trials was obtained by drying and grinding the TMR fed to donor cattle to pass through a 1-mm sieve.

The method described by Menke and Steingass [19] was used for the in-vitro gas production assay. All operations were carried out in a 39 °C water bath with a magnetic stirrer under continuous CO_2_ flushing. The artificial rumen culture solution was made by mixing a preprepared mineral buffer solution and rumen fluid (2:1). The substrates were prepared by mixing the basic fermentation substrate with appropriate amounts of PICB to obtain samples with concentrations of 0, 0.1, 0.2, 0.3, 0.4, and 0.5% PICB; each concentration was prepared in triplicate. Substrate (200 mg DM) was weighed into a 100-mL calibrated glass syringe (HFT000025, Häberle Maschinenfabrik, Germany), and 30 mL of artificial rumen culture solution was added. The syringe was kept vertical and shaken gently. By pushing the piston upward, no gas headspace at the liquid inlet end of the syringe was permitted. The silicon rubber tube was clipped, and the initial volume was recorded. Each syringe was placed in an incubator at 39 °C for 24 h, and the gas volume was recorded at different time points; the fermentation parameters and gas composition were analyzed at the end of the trial.

In-vitro gas production over 96 h was recorded at 16 different time points of fermentation using the method described above. Gas production was calculated relative to the gas volume of a blank control group at every time point. Cumulative gas production was calculated, and a gas production curve was prepared.

### 2.4. Sampling and Chemical Analyses

After 24 h of in vitro incubation, the fermentation syringes were removed and placed in an ice-water bath to terminate the fermentation. Gas production was recorded. Using a disposable syringe, 1 mL of gas was extracted from the fermentation syringe and immediately injected into a TP-2060T gas chromatograph (Beifen Tianpu Instrument Technology, Beijing, China). Chromatographic analysis was performed under the following conditions: thermal conductivity detection, TDX-01 packed column (1 m × 3 mm), inlet temperature 150 °C, column temperature 120 °C, detector temperature 150 °C, helium carrier gas (flow rate 50 mL/min), injection volume 1 mL. The CH_4_ yield was calculated from the resultant data.

Immediately after gas sampling, the pH of the fermentation liquid was measured with a pH meter equipped with a glass electrode (Model PHS-3C, Shanghai Leici Scientific Instrument, China).

To measure VFA concentration, 5.0 mL of fermentation liquid was mixed with 1 mL of phosphoric acid (25% *w/v*); the mixture was kept at 4 °C for 30 min and then centrifuged (15,000 × *g*, 15 min). The supernatant (1 mL) was used for the determination of VFA by gas chromatography (GC522, Wufeng Instruments, Shanghai, China), as described by Cao and Yang [20]. Another 4 mL of supernatant was mixed with 0.8 mL of 1% sulfuric acid for the determination of NH_3_-N using the phenol-sodium hypochlorite colorimetric method described by Broderick and Kang [21]. The samples were stored at −20 °C prior to analysis.

For the in-vitro DM degradation rate (IVDMD) assay, the substrates (600 mg DM) were prepared by mixing the basic fermentation substrate with the appropriate amount of PICB to obtain samples with concentrations of 0, 0.1, 0.2, 0.3, 0.4, and 0.5% (w/w) PICB. Each concentration was prepared in triplicate. The substrates were poured into a filter bag, placed in 100-mL glass culture tubes (in duplicate), and mixed with 70 mL of artificial rumen culture solution. The tubes were then covered quickly and placed in a 39 °C water bath with a shaker. After 24 h of fermentation, the filter bags were removed, rinsed with water, and dried at 105 °C for 12–24 h to constant weight. The in-vitro DM degradation rate (IVDMD) was calculated as follows:(1)$$\mathrm{IVDMD}\left(\%\right)=\frac{\left(W1-W0\right)}{\left(W2-W0\right)}\times 100\%$$
where W0 is the filter bag weight, W1 is the final weight of the dried sample and the filter bag, and W2 is the initial weight of the substrate and the filter bag.

### 2.5. Statistics and Analysis

The data for pH, NH_3_-N concentration, VFA, gas production, gas composition, and DM degradation were analyzed using the ANOVA of SPSS 25.0. The Tukey multi-comparison method was used to compare the effects of different doses of PICB. Linear (L) and quadratic (Q) responses were analyzed to evaluate the dose-dependent trends. The significant level was *p* < 0.05, and the highly significant level was *p* < 0.01 [22,23,24].

The nonlinear method of SPSS 25.0 was used to calculate the dynamic fermentation parameters, and an exponential model (Equation (2)) was adopted to analyze the nonlinear fitting of cumulative gas production at different PICB concentrations:*Y* = *B* [1 − *e*^(−*c*×*t*)^ ](2)
where *Y* is the cumulative gas production (mL) at time *t*, *B* is the theoretical maximum gas production (mL), *c* is the gas production rate (h^−1^), and *t* is the fermentation time (h) [25].

## 3. Results

### 3.1. Fermentation Parameters and DM Degradation Rate

Table 2 shows the fermentation parameters and the in-vitro DM degradation rate (IVDMD).

The pH (average value: 6.60 ± 0.36) was not modified by the addition of PICB, and no significant difference was observed among the different PICB concentrations (*p* > 0.05).

PICB addition significantly affected the NH_3_-N concentration during fermentation. Compared with the control group, the addition of PICB significantly reduced NH_3_-N concentration (*p* < 0.05). A PICB concentration of 0.3% showed the largest decrease, which was significantly different from the other PICB groups except for 0.2%. A significant quadratic response was observed with increasing PICB concentration (*p* < 0.01).

While the addition of PICB had no significant effect on total VFA concentration during fermentation (*p* > 0.05), the proportions of acetic, propionic, and butyric acids were significantly affected. Compared with the control group, the addition of 0.1%–0.4% PICB significantly reduced the percentage of acetic acid (*p* < 0.05), with the 0.2% and 0.3% groups significantly lower than the 0.1% and 0.4% groups. In contrast, the addition of 0.1%–0.4% PICB significantly increased the percentage of propionic acid (*p* < 0.05), with 0.2% and 0.3% groups significantly higher than the 0.1% group. Correspondingly, the addition of 0.2% and 0.3% PICB significantly reduced the acetic/propionic acid (A:P) ratio compared with the control group. The 0.3% PICB inclusion significantly increased butyric acid content during fermentation (*p* < 0.05). Highly significant quadratic responses (*p* < 0.01) were observed with the increase of PICB concentration for acetic, propionic, and butyric acids and for the A:P ratio.

PICB had no significant effect on isobutyric, isovaleric, and valeric acids and IVDMD (*p* > 0.05) at any dose.

### 3.2. Gas Production, Composition, and Methane Yield

The results related to gas production and composition after 24 h of in-vitro fermentation are reported in Table 3. Compared with the control group, 0.4% and 0.5% PICB addition significantly decreased overall gas production (*p* < 0.05); 0.2% and 0.3% PICB did not significantly influence gas production even though the values were higher. The CH_4_ percentage and volumetric yield for 0.2% and 0.3% PICB were significantly lower than those of the control group (*p* < 0.05), with the 0.2% results lower than the 0.3% results. Highly significant quadratic responses with increasing PICB concentration were observed for CH_4_ volumetric yield and CH_4_ percentage (*p* < 0.01). The addition of PICB did not significantly affect H_2_, oxygen + nitrogen, and CO_2_ content during 24 h of in-vitro fermentation (*p* > 0.05).

### 3.3. Gas Production after 96 h In-Vitro Fermentation

As shown in Table 4, the addition of PICB did not significantly affect rumen gas production parameters after 96 h of in vitro fermentation. The theoretical maximum gas production and gas production rate of PICB groups showed oscillations at about control group levels. Gas production, theoretical maximum gas production, and gas production rate showed highly significant linear responses (*p* < 0.01) with increasing PICB concentration. Gas production and maximum gas production showed highly significant quadratic responses with increasing PICB concentration (*p* < 0.01).

## 4. Discussion

### 4.1. Effect of PICB on pH

Rumen pH depends mainly on diet characteristics (chemical and physical) and can range between pH 5.6 and 7.5 in response to different feed formulations [26]. In the present study, different concentrations of PICB did not significantly affect the pH, ranging from pH 6.4 to 6.7. These findings are in agreement with those of Sarnataro and Spanghero [27] and Aguerre et al. [28], who found similar results for in-vitro fermentation and in Holstein cows. On the contrary, Hassanat and Benchaar [29] found that the addition of chestnut tannin significantly increased the pH of rumen fluid.

### 4.2. Effect of PICB on NH_3_-N Concentration

The nitrogen retention rate in the rumen is an important indicator of the efficiency of protein utilization. A higher nitrogen retention rate indicates more efficient utilization of nitrogen compounds and less nitrogen loss in urine and feces. NH_3_-N is a key nitrogen source for fermentation and one of the primary nitrogen sources for microbial growth. High NH_3_-N concentration in the rumen leads to more nitrogen loss and can even influence reproductive performance [30]. Three factors may cause a decrease of NH_3_-N content. First, Al-Dobaib [31] reported that tannins can form complexes with feed proteins and reduce their utilization by rumen microorganisms, thereby improving the nitrogen utilization rate and reducing nitrogen emissions. PICB is mainly composed of HTs such as ellagic acid, with a small percentage of CTs such as proanthocyanins [14,32]. CTs can form complexes with feed proteins and decrease their degradability [33], thus reducing the NH_3_-N concentration. HTs can also combine with protein to form HT–protein complexes, but some rumen microorganisms can dissociate the HT–protein complex [8]. Second, tannin can inhibit the activity of microbial dehydrogenase [34]. Third, the bacteria-degrading activity of protozoa can be inhibited by tannins [35]. In this study, 0.2% and 0.3% PICB decreased NH_3_-N production by 3.5% and 4.7%, respectively, which was consistent with the results of Sarnataro and Spanghero [27], who reported that the addition of 0.25%–1.5% DM chestnut tannin to beef cattle diet reduced in-vivo ammonia concentration without adverse effects on beef cattle growth [36]. The decrease of NH_3_-N concentration may be attributed to the inhibitory activity of tannin against protozoan symbiotic bacteria [35].

### 4.3. Effect of PICB on VFA Production

Carbohydrates are degraded by rumen microorganisms to form adenosine triphosphate (ATP), VFAs, CO_2_, H_2_, CH_4_, and other minor compounds. VFAs are the main source of energy for ruminants, accounting for 70%–80% of energy absorption; for this reason, the measurement of rumen VFAs is of great significance [37]. Vasta et al. [38] summarized the available literature regarding changes in levels of VFAs and acetic, propionic, and butyric acids after the use of tannin. The results varied from increased levels to unchanged or decreased levels for all four parameters, including studies using tannins from chestnut. Aboagye and Beauchemin [17] reported that the effects of tannins on intake, digestibility, rumen fermentation, CH_4_ production, and animal performance vary, depending on the part of the plant, stage of growth, and growing conditions. Therefore, the effects of tannin addition should be studied case by case. In this trial, the addition of PICB significantly affected the relative contents of acetic and propionic acids and significantly reduced the A:P ratio. The butyric acid content for 0.3% PICB was significantly higher than that of the other groups. Jayanegara et al. [22] and Bhatta et al. [8] reported similar results and suggested that certain types of tannins could improve energy efficiency. Hassanat and Benchaar [29], however, showed that the addition of chestnut tannin increased in-vitro acetic acid production and reduced the proportion of valeric acid and branched-chain VFAs without any influence on the A:P ratio. Sarnataro and Spanghero [27] reported that the addition of chestnut tannin reduced the amount of total VFAs, isobutyrate, and isovalerate and increased the proportion of valerate. However, no significant change was observed for total VFAs, isobutyrate, isovalerate, or valerate in this study.

### 4.4. Effect of PICB on Methane Production

Many authors have reported that the addition of CTs and HTs reduces CH_4_ production in the rumen [6,22]. Bhatta et al. [8] reported that chestnut tannins reduced CH_4_ production in vitro and demonstrated that the effect was associated with a reduced protozoa population. Liu et al. [39] reported that the supplement of chestnut tannins decreased CH_4_ emissions from sheep by reducing methanogen and protozoa populations. Similar results were reported by Jayanegara et al. [22], Witzig et al. [40], and Aboagye and Beauchemin [17]. Vasta et al. [38], after reviewing the microbiota analyses achieved in the past 10 years, concluded that an overall depressive effect of dietary polyphenols was related to total methanogen and protozoa populations, given that archaea are associated with protozoa and the reduced protozoa population in the presence of tannins partly contributes to decreased methane production. The decrease of methane production might be related to the transfer of the protozoan population from a type-B ciliate community to a type-A ciliate community [41]. CTs indirectly reduced the CH_4_ production by reducing fiber digestion, while HTs inhibited the activity and growth of methane- and hydrogen-producing microorganisms [6]. HT reduced CH_4_ production more efficiently than CT [22,37]. Shi [14] reported that the main effective polyphenols in PICB were HTs, such as ellagic acid and gallic acid, rather than CTs. The aim of the current study is to explore the effects of PICB on CH_4_ production. The study compared the function of PICB from *Castanea mollissima* Blume with chestnut tannins from *Castanea sativa* in studies reported by Liu et al. [39], Jayanegara et al. [22], and many others. We consider that PICB shows a similar influence on rumen microbiota.

### 4.5. Effects of PICB on IVDMD

The DM degradation rate is a reflection of ruminant diet digestion and absorption efficiency. Jayanegara et al. [22] and Jafari et al. [42] reported that HTs had no significant effect on the in-vitro DM degradation rate in ruminant diets, while CTs had more influence on DM degradation. Aboagye et al. [2,36] reported that chestnut wood tannin did not affect the IVDMD or organic matter degradation. In the current study, PICB addition did not affect IVDMD, which might be because of the gallic acid subunit or its metabolites [14,42]. However, Wischer et al. [5] reported that in-vivo DM degradation and organic matter degradation decreased after the addition of chestnut tannin to sheep diets. Further work is needed to study the effects of different sources of tannin on animal performance.

### 4.6. Effect of PICB on Gas Production after 24 and 96 h In-Vitro Fermentation

In-vitro culture gas production reflects diet degradation and microbial activity in the rumen [22]. Regarding gas production after 24 h of fermentation, the results in the literature are quite variable. Some have reported reduced gas production [29,43], some were unchanged [27], while Wei et al. [24] reported increased gas production. In the present study, the addition of 0.2% and 0.3% PICB did not significantly increase gas production when compared with the control group, even though the experimental values were marginally higher. In contrast, the addition of 0.4% PICB induced a decline in gas production and 0.5% PICB caused a significant decrease. These results were similar to those of Jayanegara et al. [22], who showed a slight decline in gas production at the 0.39% (0.5 mg/mL) addition level of chestnut tannin and a significant decrease at 0.78% (1 mg/mL).

The addition of PICB did not significantly affect 96-h gas production, although there were significant differences and highly significant linear and quadratic responses for gas production, theoretical maximum gas production (*B*), and gas production rate (*c*) during the 96-h in-vitro fermentation (see Table 4). While PICB might modify the fermentation rate, it did not modify final gas production because the fermentation bases were the same among groups. This result was in agreement with that of Sarnataro and Spanghero [27], who reported that the addition of chestnut tannin had no effect on gas production during in vitro fermentation.

## 5. Conclusions

In this study, the addition of PICB to a ruminant diet significantly affected the NH_3_-N concentration, the content of acetic, propionic, and butyric acids, the A:P ratio, and CH_4_ production after 24 h of in-vitro rumen fermentation. The addition of PICB had no influence on 24-h DM degradation or 96-h total gas production. The addition of 0.3% PICB significantly reduced the NH_3_-N concentration relative to the control, while the addition of 0.2% PICB significantly increased the relative content of propionic acid and reduced acetic acid content and the A:P ratio. Furthermore, the addition of 0.2% PICB significantly reduced the CH_4_ yield during in-vitro rumen fermentation. These results indicate that the addition of PICB is a promising dietary strategy for improving ruminant energy utilization and mitigating greenhouse gas emissions.

## Figures and Tables

**Table 1 animals-11-00108-t001:** Ingredients and characteristics of the total mixed ration.

**Ingredient g/100 g DM**	
Flaked corn	17.6
DDGS (dry) ^1^	7.8
Soybean meal	2.4
Cottonseeds fuzzy	2.9
Corn silage	66.6
Sodium chloride	0.2
Premix ^2^	2.5
**Chemical Composition % DM**	
Crude protein	13.52
NDF ^3^	31.48
ADF ^4^	16.71
Calcium (Ca)	0.81
Phosphorus (P)	0.36
Metabolized energy (MJ/kg) ^5^	11.50

^1^ DDGS, distillers’ dried grains with solubles. ^2^ Contained (per kg of premix; DM basis): 50,000 mg NaCl, 2400 mg Mg, 7600 mg K, 200 mg Cu, 400 mg Mn, 650 mg Zn, 22 mg I, 2 mg Se, 9 mg Co, 121,000 IU vitamin A, 374,000 IU vitamin D3, 5500 IU vitamin E. ^3^ NDF = neutral detergent fiber. ^4^ ADF = acid detergent fiber. ^5^ Based on NRC (2001) [18] ingredient composition.

**Table 2 animals-11-00108-t002:** pH, NH_3_-N, total VFAs and their components, and DM degradation after 24-h in-vitro fermentation with different doses of PICB^1^.

Items	PICB inclusion (DM, %)						Polynomial ^6^
	0%	0.1%	0.2%	0.3%	0.4%	0.5%	
pH	6.598 ± 0.008	6.621 ± 0.042	6.600 ± 0.066	6.596 ± 0.012	6.610 ± 0.034	6.598 ± 0.018	
NH_3_-N (mmol/L) ^2^	24.51 ± 0.76 ^a^	24.05 ± 0.62 ^b^	23.66 ± 0.58 ^bc^	23.35 ± 0.51 ^c^	23.92 ± 0.55 ^b^	23.94 ± 0.72 ^b^	Q
Total VFAs (mmol/L) ^3^	47.07 ± 1.65	46.96 ± 1.47	48.00 ± 2.32	47.42 ± 1.43	46.50 ± 2.61	48.04 ± 2.79	
Acetic acid (%)	60.38 ± 0.21 ^a^	59.98 ± 0.36 ^b^	59.03 ± 0.38 ^c^	59.02 ± 0.40 ^c^	59.82 ± 0.43 ^b^	60.22 ± 0.30 ^a^	Q
Propionic acid (%)	20.70 ± 0.39 ^d^	21.00 ± 0.31 ^c^	21.50 ± 0.41 ^a^	21.26 ± 0.41 ^ab^	21.11 ± 0.40 ^bc^	20.86 ± 0.25 ^cd^	Q
Isobutyrate acid (%)	1.35 ± 0.09	1.35 ± 0.04	1.34 ± 0.09	1.35 ± 0.06	1.34 ± 0.06	1.30 ± 0.09	
Butyric acid (%)	13.31 ± 0.37 ^b^	13.61 ± 0.33 ^b^	13.85 ± 0.34 ^ab^	14.06 ± 0.38 ^a^	13.60 ± 0.36 ^b^	13.68 ± 0.29 ^b^	Q
Isovaleric acid (%)	2.66 ± 0.07	2.70 ± 0.10	2.69 ± 0.13	2.68 ± 0.06	2.65 ± 0.07	2.64 ± 0.11	
Valeric acid (%)	1.40 ± 0.06	1.41 ± 0.04	1.42 ± 0.09	1.42 ± 0.06	1.37 ± 0.09	1.38 ± 0.06	
A:P ^4^	2.917 ± 0.054 ^a^	2.857 ± 0.037 ^a^	2.740 ± 0.056 ^c^	2.777 ± 0.062 ^bc^	2.834 ± 0.040 ^ab^	2.887 ± 0.033 ^a^	Q
IVDMD (%) ^5^	56.30 ± 0.89	56.84 ± 0.90	56.51 ± 1.12	56.14 ± 1.64	57.16 ± 2.37	56.15 ± 1.50	

Different letters indicate significant differences (*p* < 0.05). ^1^ PICB = polyphenols extracted from the involucres of *Castanea mollissima* Blume. ^2^ NH_3_-N = ammonia nitrogen. ^3^ VFAs = volatile fatty acids. ^4^ A:P = acetic acid:propionic acid. ^5^ IVDMD = in-vitro dry matter degradation after 24 h. ^6^ Q = highly significant quadratic effect of dietary addition of PICB (*p* < 0.01).

**Table 3 animals-11-00108-t003:** Gas production and composition at 24-h in-vitro fermentation with different rates of PICB addition.

Items	PICB inclusion (DM, %) ^1^						Polynomial ^2^
	0%	0.1%	0.2%	0.3%	0.4%	0.5%	
24-h gas production (mL)	48.22 ± 0.52 ^ab^	48.16 ± 0.45 ^ab^	48.53 ± 1.24 ^a^	48.23 ± 0.74 ^ab^	47.79 ± 0.68 ^bc^	47.56 ± 1.00 ^c^	
H_2_ (%)	0.0078 ± 0.0035	0.0090 ± 0.0023	0.0094 ± 0.0010	0.00823 ± 0.0042	0.0094 ± 0.0013	0.0089 ± 0.0023	
Oxygen + Nitrogen (%)	2.99 ± 0.66	2.93 ± 0.61	2.79 ± 0.63	2.74 ± 0.49	3.01 ± 0.52	2.78 ± 0.68	
CO_2_ (%)	74.66 ± 0.59	74.51 ± 1.04	74.65 ± 1.17	74.61 ± 0.87	74.24 ± 1.15	74.27 ± 1.17	
CH_4_ (%)	14.65 ± 0.27 ^a^	14.52 ± 0.27 ^a^	13.36 ± 0.35 ^c^	14.15 ± 0.47 ^b^	14.52 ± 0.38 ^a^	14.60 ± 0.45 ^a^	Q
CH_4_ yield (mL)	7.063 ± 0.101 ^a^	6.991 ± 0.109 ^a^	6.468 ± 0.171 ^c^	6.823 ± 0.214 ^b^	6.940 ± 0.249 ^ab^	6.958 ± 0.0234 ^a^	Q

Different letters indicate significant differences (*p* < 0.05). ^1^ PICB = polyphenols extracted from the involucres of *Castanea mollissima* Blume. ^2^ Q = highly significant quadratic effect of dietary addition of PICB (*p* < 0.01).

**Table 4 animals-11-00108-t004:** Effect of different concentrations of PICB on in-vitro rumen fermentation 96-h dynamic fermentation parameters.

Items	PICB Inclusion (DM, %) ^1^						Polynomial ^4^	
	0%	0.1%	0.2%	0.3%	0.4%	0.5%		
96-h gas production (mL)	61.30 ± 0.48	62.34 ± 0.92	62.52 ± 0.51	60.04 ± 0.33	60.88 ± 0.85	59.16 ± 0.91	L	Q
B (mL) ^2^	61.41 ± 0.47 ^b^	62.41 ± 0.91 ^a^	62.65 ± 0.51 ^a^	60.19 ± 0.34 ^c^	61.01 ± 0.85 ^bc^	59.29 ± 0.92 ^d^	L	Q
c (%/h^−1^) ^3^	0.0658 ± 0.0014 ^b^	0.0699 ± 0.0011 ^a^	0.0649 ± 0.0009 ^bc^	0.0630 ± 0.0015 ^d^	0.0642 ± 0.0009 ^cd^	0.0638 ± 0.0009 ^cd^	L	

Different letters indicate significant differences (*p* < 0.05). ^1^ PICB = polyphenols extracted from the involucres of *Castanea mollissima* Blume. ^2^
*B* = theoretical maximum gas production. ^3^
*c* = gas production rate. ^4^ L = highly significant linear effect of dietary addition of PICB (*p* < 0.01). Q = highly significant quadratic effect of dietary addition of PICB (*p* < 0.01).

## Data Availability

The data presented in this study are available on request from the corresponding author. The data are not publicly available due to the authors having to personal website.

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
