# Peer review of "Pilot Study of the Effects of Polyphenols from Chestnut Involucre on Methane Production, Volatile Fatty Acids, and Ammonia Concentration during In Vitro Rumen Fermentation"

_animals, 2021, doi:10.3390/ani11010108_

Round 1
Reviewer 1 Report
Dear authors,
The numerous changes, made by authors, significantly improved presented manuscript. The authors carefully reviewed the literature studies and discussed in the text [such as Vasta et al. (2019), Aboagye and Beauchemin (2019), Jafari et al. (2019), etc.], as well as answered on my major comments in details. To our regret, the authors did not take serious the remark concerning the limited number of animals (only 3) in this study. I insist on the inclusion of a complete sentence in the “Abstract” section, for example, “three healthy Angus bulls (350±50 kg), equipped with permanent rumen fistula, were used” or it short version “3 fistulated Angus bulls (350±50 kg) were used…”. It is not enough to mention this as “… n=3]”… in the following sentence “A basic diet was supplemented with five doses of PICB [0%–0.5% dry matter (DM); n =3]”. Again, it is important for the readers to evaluate the value of the obtained data and statistics.
Reviewer 2 Report
The paper does not present the concentration of tannins or which of the tannins is the cause of the effects in the reduction of Nitrogen, NH4, or which affect and as the population of protozoa according to the discussion. It is important to correct the content of Net Energy because it does not correspond to the value of any of the ingredients, it must be ME and not NEReviewer 3 Report
Comments and Suggestions for Authors
The paper “Effect of polyphenols from chestnut involucres on methane production, volatile fatty acids and ammonia concentration of in vitro rumen fermentation by Wang et al. deals with de determination of the fermentation profile of a diet for growing cattle after the addition of increasing concentrations of tannins from chestnut. The paper was submitted previously and this is a resubmission.
After a careful reading of the paper, and in spite of the efforts by authors to provide support for the value of the paper, I have major concerns that preclude possible publication. This opinion is based on the following criteria:
1.- The experiment has not been duplicated. The triplicates they use are not independent samples, because they have been conducted at the same time with same rumen fluid. In vitro studies have a large period to period variation that need to be taken into account. Replicated periods are required. Not doing so precludes any scientific validity because variability between independent samples, a basic principle of science, is not met, and stats are not valid. Under these conditions, this is not a valid scientific paper. That was observed in the previous paper, no corrections made.
2.- There are several extensive reviews on the effect of tannins (Vasta et al. 2019 in J. Dairy Sci; Aboagye and Bauchemine, 2019 in Animals; Jafari et al., Annals of Animal Science). A careful reading of these three reviews provides an overwhelming load of evidence that the research presented lacks of novelty and provides no advancement in our current knowledge on the topic. The results are fully expected and the paper does not provide any new view. Many of the papers studying tannins are more than 20 years old, and the present papers provides no other information than saying that that specific tannin also has same expected effect. At this point in science, the paper presented is just a descriptive trial that has no contribution to the advancement of science. There are no effort to search for possible mechanisms of action or demonstrate any changes in microbial profile. This is clearly reflected in the discussion, where it is limited to say “this is similar to that and different from that”, or speculate on mechanisms of actions proposed by other authors. The rest of the discussion is pure speculation based on other’s research. I see no gain in science other than a descriptive effect of a specific product. And this is not sufficient to be published. Knowing all we know about tannins already, and the hundreds of papers published in the area, a research paper should then contribute to provide explanations, not just describe an effect. I don’t see an opportunity to improve the paper (other than starting a full new research project with a deeper insight on mechanisms of action of tannins).
3.- I disagree (as indicated in the first review) in the stats approach. Authors should use only one, not two different approaches. Contrasts are more than sufficient. That was indicated in the first revision and has not been modified.
There are several other issues of concern. I will mention some (not all), but these examples reflect that the paper was not carefully planned:
Ln 114. The substrate of fermentation is not described. This is very important.
In ln 111. It is indicated that samples are mixed with a magnetic stirrer, but incubations occur in syringes (line 116), How can this be possible?
Ln 115. The % content of treatments…. Percent of what? Of the total volume? Of the substrate of fermentation? Very confusing.
Ln 144-151: This is not Sampling and Chemical Analysis: reflects poor manuscript preparation. The IVDMD was not replicated either!!
Ln 147: The filter bags are not described. The pore size of bags is very important. It should allow the transit of bacteria. Many filter bags are too small or too large pore sizes. That affects IVDMD.
Ln 168: You do not measure rate of IVDMD. You measure extent of degradation. Reflects poor understanding.
Ln 122: Why are two gas production conducted? What is the purpose? If you incuabate 96 h, you already have the 24 measurement. It makes no sense!!. reflects poor planning.
Table 2. This is not the way of reporting results from ANOVA. A model SEM should be reported. It reflects poor understanding of statistics.
The discussion is full of speculation and incorrect statements: Some examples.
line 232 : “improving N utilization…” reducing N degradation in the rumen does not imply improved use of N. Could be indigested in the small intestine or not required by the animal and, therefore, excreted.
ln 241: The potential action of tannins on microbes has not been demonstrated by authors. A lower ammonia may be due to lower protein degradation by direct binding. Reflects poor discussion.
Ln 265: All this discussion, like most of the discussion section, is a review of other’s findings, but nothing that is added from the current research. Authors have not made any effort to provide additional information on how tannins work.
I have other comments, but I think that there are several major problems in the paper (lack of novelty, lack of replication, small in scope and objective of limited value, only descriptive, no added knowledge in science, etc, ) that preclude the possible publication of the paper.
This manuscript is a resubmission of an earlier submission. The following is a list of the peer review reports and author responses from that submission.
Round 1
Reviewer 1 Report
Dear authors,
The presented manuscript is an important and actual. The manuscript is devoted to the investigation of the effects of polyphenols extracted from involucres of Castanea mollissima Blume (i.e. PICB) on in vitro rumen fermentation. The manuscript analyze the literature works in detail and at relatively high level of discussion. It is positive that numerous parameters have been measured by the authors: rumen pH, the volatile fatty acids (VFA) and ammonia nitrogen (NH3-N) concentrations, methane (CH4) yield and DM degradation rate (IVDMD) - after 24 hours in vitro fermentation, as well as the gas production - monitored for 96 hours. It is positive that linear (L) and quadratic (Q) responses have been analyzed by the authors “to evaluate the dose-dependent trends”, as well as a non-linear model and an exponential model have been used “to calculate the dynamic fermentation parameters and the nonlinear fitting of cumulative gas production at different PICB concentrations”. The authors declared that #1) the addition of 0.3% PICB significantly reduced the NH3-N concentration (as compared to the control group) and a linear response with increasing PICB concentration was observed; #2) the addition of 0.2% and 0.3% PICB significantly increased the content of propionic acid, reduced the acetic acid content and the acetic/propionic (A:P) ratio; #3) the addition of 0.2% and 0.3% PICB significantly reduced the CH4 production, etc.
But the only point #1) is more or less acceptable - the addition of 0.3-0.5% PICB reduced the NH3-N concentration at 8.0-6.5%, respectively (as compared to the control group). In contrast, the point #2) has a lot of contradictions: the addition of 0.2% and 0.3% PICB increased the content of propionic acid at 5.36% or 3.65%, as well as reduced the acetic acid content only at 2.39% or 2.38%, i.e. insignificantly, to my opinion. The point #3) is not acceptable, because the addition of 0.2%, or 0.3% PICB increased the CH4 production at 24 h by 3.8% or 1.68% only, whereas the addition of 5% PICB reduced the CH4 production at 24 h by “-“2.66% (Table 3) ! Moreover, the addition of 0.2%, 0.3% or 0.5% PICB reduced the CH4 % at 24 h by 10.0%, 4.2% or 0% (Table 3) ?! This effect is strange and has no reasonable explanation in the text. Just think – 10% reduction after 0.2% PICB addition ? At the same table 3 - 0% reduction (i.e. unchanged values) after 0.5% PICB addition ? The authors found completely different dependences (by PICB addition) for the same parameters after 78 h (table 4). These effects have no reasonable explanation in the text.
I do not doubt the technical quality of the work, but feel that there is not sufficient data presented (just 3 bulls “equipped with permanent rumen fistula, were selected as rumen fluid donors”) and result discussion. It is necessary to consider a major revision of this manuscript to provide the highest impact on a broader readership to justify publication in the "Animal". This topic is in the frame of the journal scope, but the subject matter is not treated in depth.
Reviewer 2 Report
Correct: The net energy value is wrong.
To review the statistics there are some data in which the values may be similar.
The gas equation is poorly expressed, it is exponential and must be expressed B (1-e ^ (- cxt))
The results in fatty acids and gas production are the effect of modification in the microbial population which was not clear, it seems that the concentration changed only by the addition, so the discussion should focus more on the effect of phenolic compounds in the CHESNUT in which type of microbial populations, has a very interesting data in which it increases the production of gas but the concentration of methane is reduced, and they do not discuss it.
Reviewer 3 Report
The paper “Effect of polyphenols from chestnut involucres on methane production, volatile fatty acids and ammonia concentration of in vitro rumen fermentation by Wang et al. deals with de determination of the fermentation profile of a diet for growing cattle after the addition of increasing concentrations of tannins from chestnut. The topic, in general terms, falls within the area of interest of readers of Animals.
The paper is simple and with a very short scope. The experimental design is short (not even replicated periods), exploring a small range of effects (VFA, methane and NH3) to be an in vitro study. And it only deals with a general extraction of chestnut tannins. There has been a very large and extensive research on the effects of tannins en general, and on their effects on ammonia reduction and methane emissions in recent years. The results of the trial are fully predictable from current knowledge and I do not appreciate any advancement in knowledge. Furthermore, if the objective would be attractive, this type of research would require to test more product or explore other potential effects that would help to explain mechanisms of action, or describe why do tannins are making these effects. Some examples (but there are many others), or full reviews on the effects of tannins (including chestnuts), have been published very recently (Vasta et al. 2019 in J. Dairy Sci; Aboagye and Bauchemine, 2019 in Animals; Jafari et al., Annals of Animal Science). A careful reading of these three reviews provides an overwhelming load of evidence that the research presented lacks of novelty and provides no advancement in our current knowledge on the topic. The results are fully expected and the paper does not provide any new views
Furthermore, the paper requires a very careful review of writing style and grammar. The stats section is also very confusing because authors used two different approaches. Authors should decide to use one or another method for the analysis (polynomial contrasts would be my choice)
In that situation, I think that it is not necessary to go into a detailed revision of the paper.